

# Use of conditioned media (CM) and xeno-free serum substitute on human adipose-derived stem cells (ADSCs) differentiation into urothelial-like cells

Ban Al- kurdi[1,4], Nidaa A. Ababneh[1], Nizar Abuharfeil[2], Saddam Al Demour[3] and Abdalla S. Awidi[1,4]

[1] Cell Therapy Center, University of Jordan, Amman, Jordan

[2] Department of Biotechnology and Genetic Engineering, Jordan University of Science and Technology, Irbid, Jordan

[3] Department of Urology, School of medicine, University of Jordan, Amman, Jordan, University of Jordan, Amman, Jordan

[4] Department of Hematology and Oncology, Jordan University Hospital, Amman, Jordan

Corresponding author
Abdalla S. Awidi,
abdalla.awidi@gmail.com

## ABSTRACT

**Background**. Congenital abnormalities, cancers as well as injuries can cause irreversible damage to the urinary tract, which eventually requires tissue reconstruction. Smooth muscle cells, endothelial cells, and urothelial cells are the major cell types required for the reconstruction of lower urinary tract. Adult stem cells represent an accessible source of unlimited repertoire of untransformed cells.

**Aim**. Fetal bovine serum (FBS) is the most vital supplement in the culture media used for cellular proliferation and differentiation. However, due to the increasing interest in manufacturing xeno-free stem cell-based cellular products, optimizing the composition of the culture media and the serum-type used is of paramount importance. In this study, the effects of FBS and pooled human platelet (pHPL) lysate were assessed on the capacity of human adipose-derived stem cells (ADSCs) to differentiate into urothelial-like cells. Also, we aimed to compare the ability of both conditioned media (CM) and unconditioned urothelial cell media (UCM) to induce urothelial differentiation of ADCS in vitro.

**Methods**. ADSCs were isolated from human lipoaspirates and characterized by flow cytometry for their ability to express the most common mesenchymal stem cell (MSCs) markers. The differentiation potential was also assessed by differentiating them into osteogenic and adipogenic cell lineages. To evaluate the capacity of ADSCs to differentiate towards the urothelial-like lineage, cells were cultured with either CM or UCM, supplemented with either 5% pHPL, 2.5% pHPL or 10% FBS. After 14 days of induction, cells were utilized for gene expression and immunofluorescence analysis.

**Results**. ADSCs cultured in CM and supplemented with FBS exhibited the highest upregulation levels of the urothelial cell markers; cytokeratin-18 (CK-18), cytokeratin-19 (CK-19), and Uroplakin-2 (UPK-2), with a 6.7, 4.2- and a 2-folds increase in gene expression, respectively. Meanwhile, the use of CM supplemented with either 5% pHPL or 2.5% pHPL, and UCM supplemented with either 5% pHPL or 2.5% pHPL showed low expression levels of CK-18 and CK-19 and no upregulation of UPK-2 level was observed. In contrast, the use of UCM with FBS has increased the levels of CK-18 and CK-19, however to a lesser extent compared to CM. At the cellular level, CK-18 and

UPK-2 were only detected in CM/FBS supplemented group. Growth factor analysis revealed an increase in the expression levels of EGF, VEGF and PDGF in all of the differentiated groups.

**Conclusion**. Efficient ADSCs urothelial differentiation is dependent on the use of conditioned media. The presence of high concentrations of proliferation-inducing growth factors present in the pHPL reduces the efficiency of ADSCs differentiation towards the urothelial lineage. Additionally, the increase in EGF, VEGF and PDGF during the differentiation implicates them in the mechanism of urothelial cell differentiation.

## INTRODUCTION

Congenital abnormalities, cancers as well as injuries may lead to irreversible damage to the urinary tract, which eventually requires reconstruction. Early attempts on the reconstruction of urinary tract have focused on the use of acellular matrix for bladder reconstruction. Acellular matrices are immunologically inert and act in vivo as scaffolds, to recruit the progenitor cells and infiltrate the matrix to produce bladder tissue (*Howard et al., 2008*). The two most commonly used acellular matrices for bladder and urethral reconstruction are the small intestinal submucosa (SIS) and the bladder acellular matrix (BAM) (*Staack et al., 2005*). In the year 2000, the FDA approved the use of porcine collagen matrix, derived from the small intestinal submucosa, in the reconstruction-based surgical procedures (*Hodde & Johnson, 2007*). However, fibrosis, matrix shrinkage and the lack of the ability to perform urothelial anastomosis are the major hurdles that must be overcome before acellular matrices can be used in the bladder and urethral reconstruction (*Portis et al., 2000*; *Campodonico et al., 2004*; *Azadzoi et al., 1999*; *Horst et al., 2019*).

Adult stem cells represent an accessible source of unlimited repertoire of untransformed cells. Early attempts to incorporate stem cells in urinary tract tissue engineering culminated in using stem cells without transdifferentaition and directly implanting them in vivo (*Liao et al., 2013*; *Chung et al., 2005*). Mesenchymal stem cells (MSCs) have been used in organ reconstruction by expanding them in vitro and then implanting them to induce their differentiation potential (*Sharma et al., 2010*; *Sharma et al., 2009*). On the other hand, stem cells can be expanded and differentiated in vitro and then transplanted directly to the affected subject. Thus, direct differentiation of stem cells can reduce the time required for patient's recovery.

Smooth muscle cells, endothelial cells, and urothelial cells are the major cell types required for the reconstruction of lower urinary tract (*Qin et al., 2014*). Three induction protocols have been reported to induce the differentiation of stem cells towards the urothelial lineage: direct co-culture with urothelial cells, indirect co-culture system with urothelial cells, and culturing the stem cells in conditioned media (CM) derived from urothelial cell culture (*Becker & Jakse, 2007*). The first protocol of direct co-culture system

is inapplicable in cases of malignancies, infections, and inflammatory diseases (*Liu et al., 2009*; *Shi et al., 2012*). The indirect co-culturing is applicable in small systems such as the filter well insert (*Liu et al., 2009*). To the current time, the use of conditioned media represents the most favorable and easy-to-use method for induction of MSCs to urothelial cells. However, several problems still need to be addressed before using these cells in clinical therapy, including the limited differentiation capability (transdifferentaition efficiency ranges from 40–70%) and the presence of xenogeneic substances such as fetal bovine serum (FBS) in cell culture media (*Zhang et al., 2014*; *Shi et al., 2012*).

Fetal bovine serum is the most vital supplement used in cell culture media for cell proliferation and differentiation. However, due to its limited supply and the increased demand on manufacturing xeno-free stem cell-based cellular products, optimizing the composition of culture media and the type of serum used are of critical importance.

Human platelet lysate (HPL) is defined by Marx as "the volume of plasma fraction of autologous blood having a platelet concentration above baseline". It has been reported that HPL enhances the proliferation and differentiation of MSCs compared to xenogenic FBS (*Kakudo et al., 2008*; *Lucarelli et al., 2003*; *Li et al., 2013*; *Mishra et al., 2009*; *Cervellia et al., 2012*). These effects make the HPL as an attractive alternative that can be used in MSCs culture with minimal adverse effects in clinical settings. In this study, the effects of FBS and pooled human platelet (pHPL) lysate were assessed on the capacity of huADSCs to differentiate into urothelial-like cells. Also, we aimed to compare the ability of both conditioned media (CM) and unconditioned urothelial cell media (UCM) to induce urothelial differentiation of ADCS.

## METHODS

### Material and reagents

Dulbecco's Modified Eagle's Medium DMEM (GIBCO, Waltham, MA, USA), collagenase type l (Worthington, Lakewood, NJ, USA), FBS (GIBCO, Waltham, MA, USA), streptomycin and penicillin and 20 mM L-glutamine (Euroclone, Italy), 0.25% trypsin–0.04% EDTA (GIBCO, Waltham, MA, USA), (SV-HUC-1) ATCC (CRL-9520), StemPro Adipogenesis differentiation media (Invitrogen, Hercules, CA, USA), Trizol reagent (Invitrogen, Hercules, CA, USA), Human MSC Analysis Kit (BD, Franklin Lakes, NJ, USA), iScript reverse transcription supermix (BioRad, Hercules, CA, USA), iQTM SYBR mix (BioRad, Hercules, CA, USA), cytokeratin-18 (Abcam, ab668, 1:200), uroplakin-2 (Santa Cruz, sc-15178, 1:50), ELISA kits (Abcam, UK).

### Cell culture

All experimental protocols involving human tissues were approved by the Ethics Committee at the King Abdullah Hospital, School of Medicine, Jordan University of Science and Technology (IRB No: IRB/7/2019). After obtaining signed informed consents, human adipose tissue aspirates were obtained from three individuals, aged 30, 31 and 35, who underwent liposuction procedures. ADSCs were isolated as previously described (*Francis et al., 2010*). Briefly, adipose tissue aspirates were digested with 0.1% collagenase type l (Worthington, Lakewood, NJ, USA) in PBS, for 45 min at 37 °C, with gentle shaking

every five minutes. The enzyme was then diluted with an equal amount of complete cell culture medium consisting of DMEM (GIBCO, Waltham, MA, USA) supplemented with 10% FBS (GIBCO, Waltham, MA, USA), 1% streptomycin and penicillin and 20 mM L-glutamine (Euroclone, Italy). The suspension was centrifuged at $1200\times$ g for 10 min and then the pellet was resuspended in 5 ml complete cell culture medium. After that, the cell suspension was passed through a 70 $\mu$m cell strainer and centrifuged at $500\times$ g for 10 min. The obtained cells were counted and seeded at a density of $2\times10^5$ cells/cm$^2$ in a tissue culture flask and incubated at 37 °C and 5% $CO_2$. The medium was changed every two to three days until the adherent ADSCs became 70–80% confluent. Cells were detached with 0.25% trypsin–0.04% EDTA (GIBCO) solution, and the resulting ADSCs at passage 3–5 were used for further experiments. SV40 immortalized human ureter urothelium (SV-HUC-1) cell line was obtained from ATCC (CRL-9520). SV-HUC-1 cells were cultured in F-12K medium supplemented with 10% FBS and 1% streptomycin and penicillin. Conditioned medium derived from SV-HUC1 was collected and used to induce the ADSCs differentiation towards the urothelial-like cell lineage.

## Flow cytometry

Cultured ADSCs at passage 3 and 70% confluency were utilized for cell surface marker assessment. Cells were detached using TrypLE (GIBCO) and washed twice with FACS buffer (PBS, 1% FBS). Then, cells were counted and adjusted to $10^6$ cells/ml. Aliquots of 100$\mu$l were placed in test tubes and incubated with fluorochrome-conjugated antibodies using Human MSC Analysis Kit (BD, USA), which includes: CD-44, CD-105, CD-73, CD-90 and a negative cocktail includes: CD-34, CD-11b, CD-19, CD-45 and HLA-DR mix for 30 min in the dark, according to the manufacturer's instructions. Cells were then centrifuged at $300\times$ g for 5 min and resuspended in 500 $\mu$l FACS buffer. The analysis was performed using BD FACSCanto$^{TM}$ and the data were analyzed using Diva software.

## Multilineage differentiation

Adipogenic differentiation was performed using StemPro Adipogenesis differentiation media (Invitrogen, Carlsbad, CA, USA) for 14 days. After that, cells were washed twice with PBS, fixed in 4% formaldehyde for 15 min and stained with oil red O stain, to confirm the presence of adipocytes. StemPro Osteogenic differentiation kit (Invitrogen) was used to induce ADSCs differentiation towards the osteogenic lineage. After 21 days in culture, differentiated cells were washed, fixed in 4% formaldehyde for 15 min, and stained with Alizarin Red S (ARS) stain, to verify the osteogenic differentiation. Cells under normal culture conditions were used as negative controls.

## Preparation of pooled human platelet lysate (pHPL)

Platelet bags designated as platelet-rich plasma 1 (PRP1) were obtained from Jordan University Hospital/blood bank unit. Briefly, platelet bags from 17 donors were pooled in one container and centrifuged at $700\times$ g for 17 min at 18 °C. After centrifugation, the platelets pellet was formed and the supernatant was designated as platelet-poor plasma (PPP). The latter was transferred into new sterile tubes, and platelets obtained from 1ml PRP1 were resuspended in 300$\mu$l of PPP, this was designated as PRP2. Following that,

**Table 1** qPCR primer sequences.

| Gene | Forward Primer | Reverse Primer |
|------|----------------|----------------|
| Uroplakin-2 | CGGAGAGCCGACAGCAAAC | ACTGAGCCGAGTGACTGTGAAG |
| Cytokeratin-18 | GGTCAGAGACTGGAGCCATTA | GGCATTGTCCACAGTATTTGC |
| Cytokeratin-19 | CGGGACAAGATTCTTGGT | CCTTGATGTCGGCCTCCA |
| GAPDH | CAAGGTTGACAACTTTG | GGGCCATCCACAGTCTTCTG |

platelets concentration was adjusted to $2\times10^6$ platelets/µl with PPP, and lysed through two freeze/thaw cycles at −20 °C and then at 37 °C. Platelet fragments were removed by centrifugation at 3000× g for 20 min at 18 °C and filtrated through a 0.2 µm filter. The obtained supernatant is now called pHPL. The pHPL was aliquoted and stored at −20 ° C. Upon supplementing the media with 10% pHPL, 2 IU/mL of heparin was added to prevent coagulation.

## Differentiation of ADSCs into urothelial-like cells

When SV-HUC1 cells cultured in F-12K medium reached 80–90% confluency, medium was collected every 24 h and changed into fresh medium for two sequential days. Next, collected medium was centrifuged at 3000 rpm for 5 min, pooled and filtered through a 0.22 µm filter and stored at −20 °C. For urothelial induction, ADSCs were seeded at a density of 500 cells/cm² andcultured with either conditioned or unconditioned medium (UCM) at a ratio of 1:4 F-12K: DMEM, supplemented with either 5% pHPL, 2.5% pHPL or 10% FBS. Uninduced controls were also included. On day 14 post-induction, cells were utilized for immunofluorescence staining and RNA extraction for relative gene expression analysis.

## qRT-PCR gene expression analysis of urothelial markers

Total RNA was isolated at day 14 of urothelial differentiation using Trizol reagent (Invitrogen, USA) and subsequently purified with RNeasy mini kit (Qiagen, Germantown, MD, USA). cDNA was synthesized using iScript reverse transcription supermix (BioRad, Hercules, CA, USA). Quantitative RT-PCR (qPCR) reactions were performed using iQ™ SYBR mix (BioRad, Hercules, CA, USA) and 300nM of each forward and reverse primers. Primer sequence and product size are provided in Table 1.

## Immunofluorescence staining of Urothelial Markers

After 14 days of induction, cells on coverslips were fixed in 4% formaldehyde for 15 min and permeabilized with PBS/0.1% Triton X-100 for 5 min. To prevent nonspecific binding, cells were incubated with blocking solution (3% BSA (wt/v) and 0.3% Triton X-100 (v/v) in PBS) for 60 min. Cells were then incubated with the primary antibodies against cytokeratin-18 (Abcam, ab668, 1:200) and uroplakin-2 (Santa Cruz, sc-15178, 1:50), diluted in blocking buffer overnight at 4 °C in a humid chamber. Subsequently, cells were incubated with the appropriate secondary antibodies, either chicken anti-mouse IgG –FITC or donkey anti-Goat-IgG Cy3, for 1 h at room temperature followed by counterstaining with DAPI (4 ′,6-diamidino-2-phenylindole) and mounting with mounting media (Invitrogen).

## Enzyme linked immunosorbent assay (ELISA)

The secretion of epidermal growth factor (EGF), vascular endothelial growth factor (VEGF), and platelet-derived growth factor (PDGF-BB) in conditioned and unconditioned culture medium was measured using ELISA kits (Abcam, Cambridge, UK), according to the manufacturer's instructions. For growth factors measurement, fresh serum-free media was added on day 14 of urothelial induction and collected after 24 h. Triplicate samples were run in 96-well plates coated with an antibody specific to a particular growth factor mentioned above. The absorbance was measured at 450 nm and within 30 min of completing the assay.

## Statistical analysis

All the experiments were performed at least three times, and statistical analysis was performed using SPSS 20.0. The data were represented as the mean $\pm$ standard error of the mean (SEM) and tested for normality and equal variance before analysis using the Shapiro–Wilk test. Statistical differences were calculated using analysis of variance (ANOVA) and Post-hoc test for comparison between groups. The analysis of ELISA data was performed using Graphpad Prism for curve fitting and independent $t$-test for significance calculation. Differences were considered significant at $P < 0.05$.

# RESULTS

## Isolation and Characterization of ADSCs

Cells with fibroblastic morphology were adhered to the tissue culture plate and reached the confluency within two weeks of the initial plating (Fig. 1A). Flow cytometry staining of the most common MSC markers showed positive expression of the following markers: CD-44 (100%), CD-105 (89.8%), CD-73 (99.9%), and CD-90 (100%) (Figs. 1B–1E). To confirm the purity of the isolated ADSCs from the hematopoietic stem cell contamination, flow cytometry staining was performed and demonstrated minimal expression levels of the negative cocktail markers (Fig. 1F). To further validate the stemness of the isolated cells, ADSCs were transdifferentiated into the osteogenic and adipogenic cell lineages. Following 21 days of the osteogenic induction, cells exhibited flattened and more elongated morphology with extracellular calcium phosphate deposits as confirmed by Alizarin red S (ARS) staining. These deposits were absent in the uninduced ADSCs cultured in cell culture media (Figs. 2A–2D). Additionally, adipogenic differentiation showed intracellular localization of lipid droplets. These droplets were positively stained with oil red O, and were absent in the uninduced negative control. Thus, the successful differentiation of ADSCs into osteoblasts and pre-adipocytes confirmed the multipotency of these cells (Figs. 2E–2H).

## Urothelial cell markers detection by qRT-PCR

To evaluate the capacity of ADSCs to differentiate towards the urothelial-like lineage, cells were cultured with either conditioned (CM) or unconditioned urothelial cell media (UCM), supplemented with either 5% pooled human platelet lysate (pHPL), 2.5% pHPL or 10% FBS. After 14 days of induction, cells were utilized for further experiments. To analyze

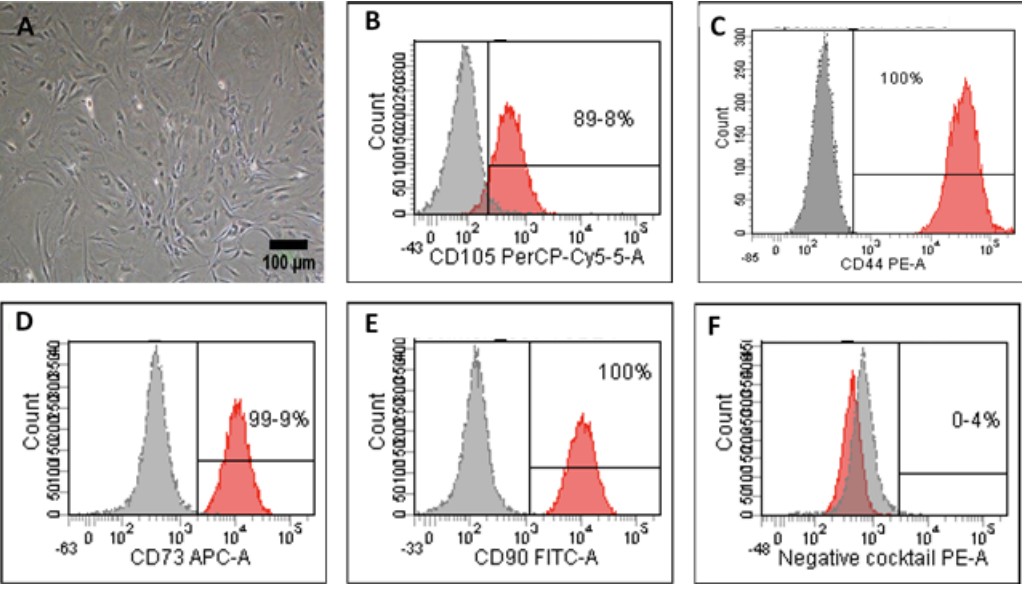

**Figure 1 Characterization and differentiation potential of ADSCs.** Characterization and differentiation potential of ADSCs. (A) Primary ADSCs morphology after 14 days in culture under the inverted phase contrast microscope. Scale bar = 100 μm. (B–F) Flow cytometry staining of ADSC markers. Cells showed positive staining for mesenchymal stem cells markers CD-44, CD-105, CD-73, CD-90 and negative for CD-34, CD-11b, CD-19, CD-45 and HLA-DR in the negative cocktail.

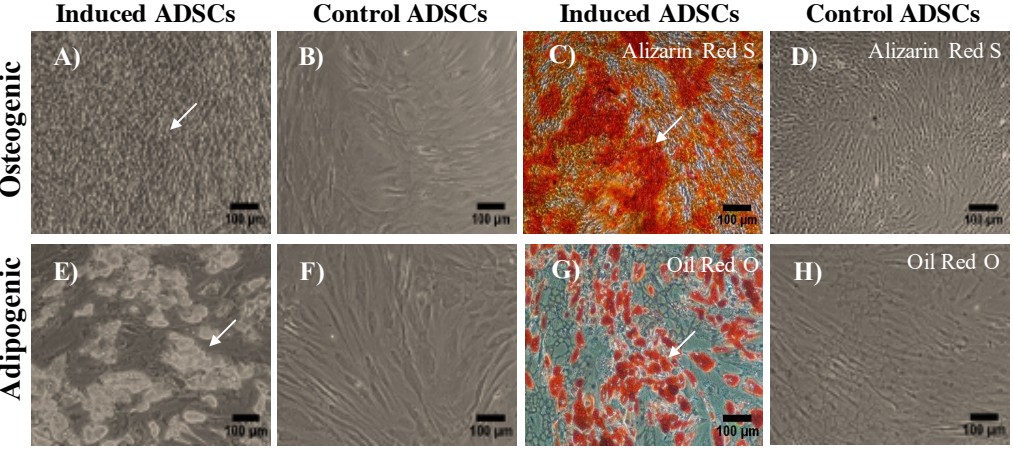

**Figure 2 Multilineage differentiation potential of ADSCs.** (A–C) Osteogenic mineral deposition was observed after 21 days of osteogenic induction and positively stained with Alizarin Red S stain (ARS). (B–D) Uninduced ADSCs were used as a negative control and stained negatively with ARZ. (E–G) Lipid droplets were observed after 14 days of adipogenic differentiation of ADSCs and positively stained with Oil Red O. (F–H) ADSCs with normal culture media stained negatively for oil red O. Scale bar = 100 μm. All differentiation experiments were repeated at least three times.

the effect of induction CM and UCM media on the differentiation of ADSCs towards the urothelial lineage, we measured the gene expression levels of two cytokeratin proteins;

CK-18 and CK-19 and one uroplakin protein (UPK-2), expressed by the urothelial cells (Fig. 3). CK-18 and CK-19 are considered as early markers of urothelial cell specification, and uroplakin proteins representing the terminal maturation stage. The levels of CK-18 expression were not significantly altered in groups treated with CM supplemented with either 5% or 2.5% pHPL ($P > 0.05$). Meanwhile, a 6.7-fold upregulation in CK-18 expression compared to the uninduced control was observed in CM FBS treated group ($P = 0.006$) (Fig. 3A). Additionally, UCM increased CK-18 expression of 3-fold in the presence of either FBS or pHPL relative to the uninduced cells (P-Value). On the other hand, culturing ADSCs in CM supplemented with FBS exhibited the highest upregulation level of CK-19 with a 4.2-fold increase compared to the uninduced control ($P = 0.01$). In contrast, CM 2.5% pHPL and CM 5% pHPL failed to upregulate the expression of CK-19. Whereas, UCM-FBS induced CK-19 expression, but to a lesser extent compared to CM-FBS. On the contrary, UCM supplemented with 5% and 2.5% pHPL failed to upregulate the expression of CK-19 (Fig. 3B). Regarding the UPK-2 terminal differentiation marker, only CM-FBS and UCM-FBS cultures showed an increased level of expression of approximately 2-fold compared to control cells (Fig. 3C).

## Detection of urothelial cell markers by immunofluorescence

Since gene expression results suggest an enhanced cellular differentiation using CM culture conditions, we compared the expression of CK-18 (early differentiation marker) and UPK-2 (late differentiation marker) between pHPL and FBS supplemented cultures by immunofluorescence staining (Fig. 4). Staining revealed that CM supplemented with FBS resulted in a 2.5-fold increase in the expression of CK-18 early differentiation marker and a 2-fold increase in UPK-2 expression compared to non-induced ADSCs control (Figs. 4A & 4B). Whereas groups treated with CM & 5% pHPL or 2.5% pHPL failed to elicit the same response (Figs. 4A & 4C).

## Assessment of growth factor levels of induced ADSCs by ELISA

To assess changes in the growth factor levels produced by the induced cells, we measured the levels of EGF, PDGF-BB, and VEGF, the main growth factor proteins secreted by cells into the culture medium. Following 14 days of urothelial induction, measurement of the growth factors in serum-free media collected after 24 h resulted in the detection of higher levels of these factors in cells treated with CM compared to their counterparts cultured in UCM (Fig. 5). Additionally, cells cultured in 5% pHPL produced higher levels of EGF, PDGF-BB, and VEGF. Levels of VEGF were significantly higher in CM-5% pHPL, CM-2.5% pHPL and UCM-pHPL ($P = 0.009, 0.023, 0.004$, respectively). Cells in FBS containing media showed the least amount of secretion of all three growth factors. Whereas, CM-FBS elicited higher levels of growth factors compared to UCM-FBS with 1.4-fold, 2.5-fold, and 2.6-fold difference for EGF, PDGF-BB, and VEGF, respectively (Fig. 5).

## DISCUSSION

Currently, cell-based therapy and tissue engineering studies mostly rely on the ex-vivo expansion and differentiation of many cell types especially stem cells. The preferential

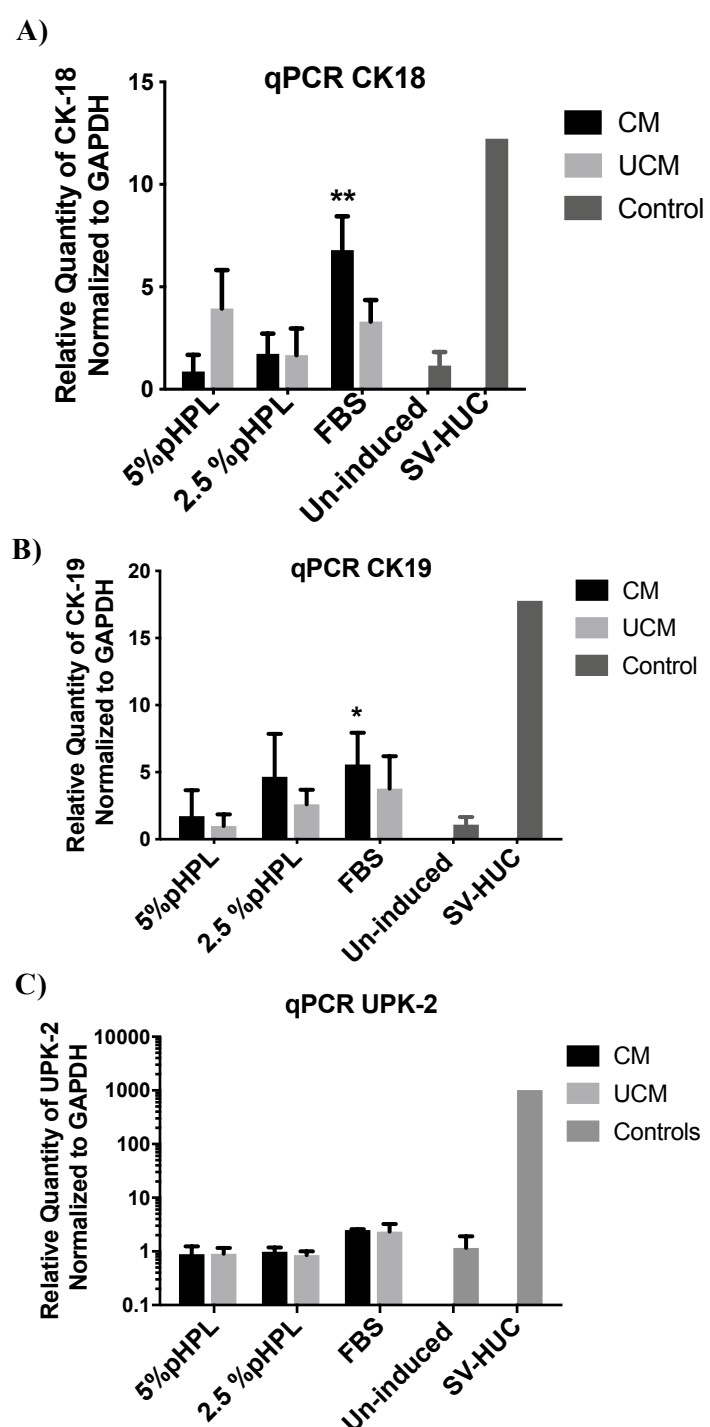

**Figure 3** **Relative gene expression of urothelial markers.** Real time-PCR performed on urothelial-like cells differentiated from ADSCs, cultivated in either conditioned medium (CM) or unconditioned urothelial cell media (UCM) in the presence of either 5% pHPL, 2.5% pHPL or 10% FBS. Uninduced ADSCs were used as the calibrator sample. (A) Relative expression of cytokeratin-18, (B) cytokeratin-19 and (C) uroplakin-2. * $P < 0.05$, ** $P < 0.01$. All experiments were repeated at least three times.

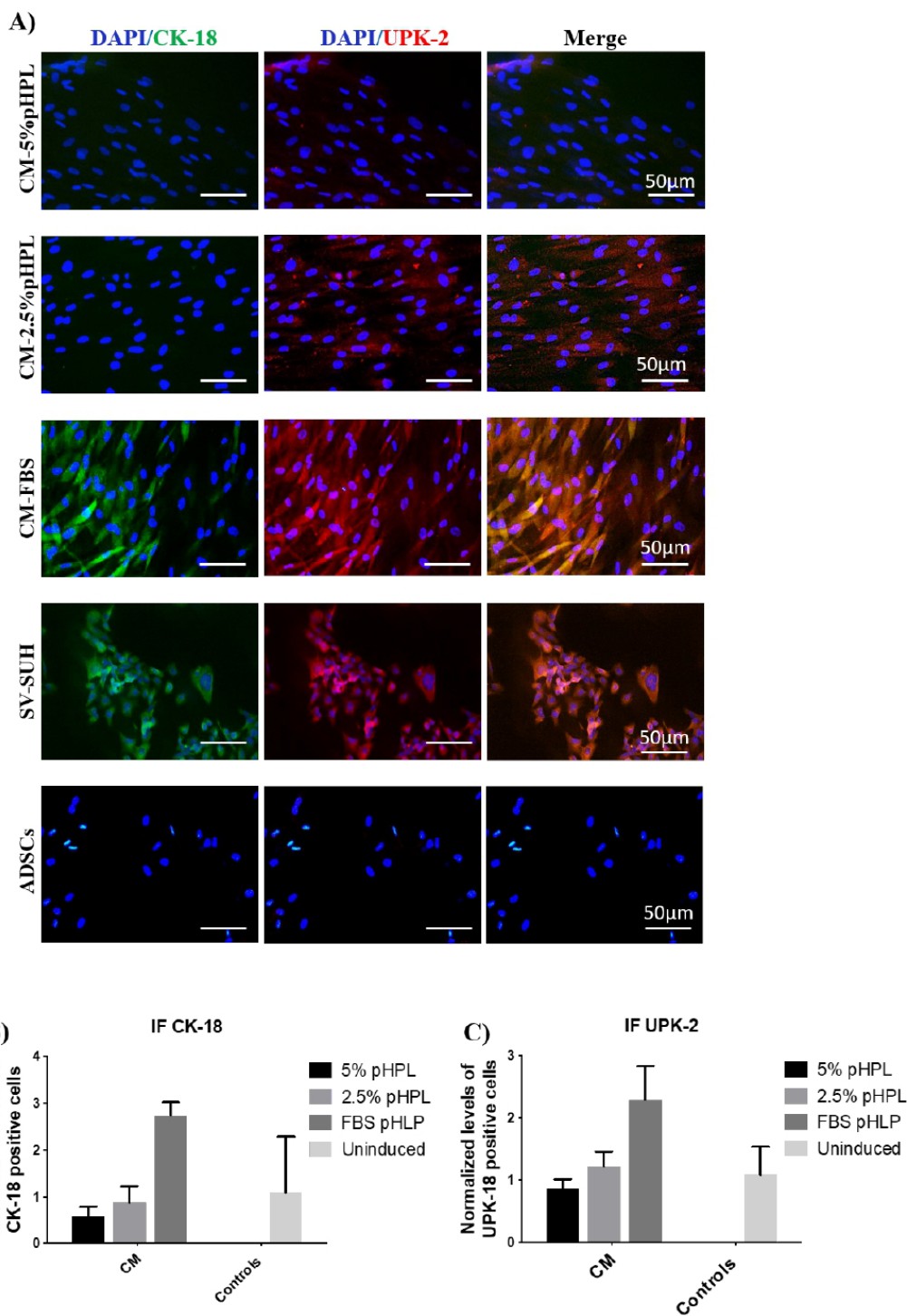

**Figure 4** **Immunofluorescence staining of urothelial markers at the cellular level.** (A) ADSCs cultured in CM and different concentrations of pHPL (5% & 2.5%) or with FBS, were assessed for the expression of cytokeratin-18 (FITC, green) and uroplakin- 2 (Cy3, red). SV-HUC cells were used as a positive control, meanwhile ADSCs were utilized as a negative control. Scale bar = 50 μm. (B & C) Semi-quantitative analysis of immunoflouresence representing percentage of positive cells relative to negative control. All experiments were repeated at least three times.

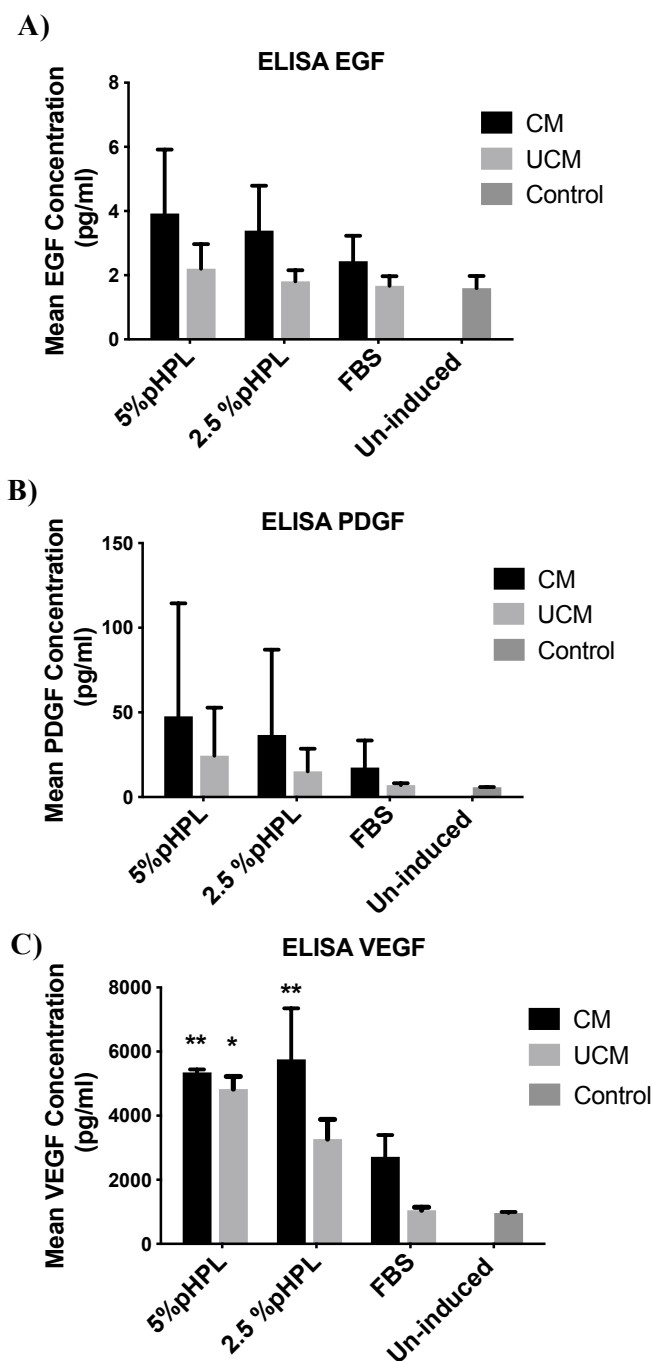

**Figure 5** **Growth factor levels assessment in induced ADSCs.** Enzyme linked immunosorbent assay (ELISA) was preformed on conditioned media collected after 24 h from cells induced for 14 days with urothelial cell derived conditioned medium (CM) or unconditioned urothelial cell media (UCM) with 5% pHPL, 2.5%pHPL or FBS. (A) Measurement of epidermal growth factor levels (EGF). (B) Platelet derived growth factor-BB (PDGF-BB). (C) Vascular endothelial growth factor (VEGF). * $P < 0.05$, ** $P < 0.01$. All experiments were repeated at least three times.

use of MSCs over other types of stem cells is related to their ability to cross lineage commitment, they differentiate efficiently into many cell types, and their inability to form teratomas or tumors in vivo (*Qin et al., 2014*). Furthermore, MSCs exhibit low immunogenicity and possess immunosuppressive capabilities, facilitating their use in allogenic stem cell transplantation studies (*Klyushnenkova et al., 2005*; *Le Blanc et al., 2003*). However, the clinical use of MSCs is hampered by their limited availability, growth variability, invasive collection procedures, and the use of xenogeneic sources of serum such as FBS during the expansion and differentiation procedures. Conditioned medium is widely used to induce MSC differentiation into chondrocytes, osteocytes, dopaminergic neurons, cholinergic neurons as well as urothelial cells (*Alves da Silva et al., 2015*; *Heino, Hentunen & Väänänen, 2004*; *Aliaghaei et al., 2016*; *Borkowska et al., 2015*; *Zhang et al., 2014*).

Different types of MSCs have been differentiated efficiently towards urothelial cells using urothelial cell-derived CM or FBS, making them an attractive source for urinary tract tissue regeneration (*Shi et al., 2012*; *Zhang et al., 2013*). However, a cell line-derived CM might contain many undefined factors, which could influence the cells in a myriad of ways. In addition, the use of FBS carries the risk of infectious disease transmission and immunization, which results in restricting the use of such system in a clinical setting (*Heiskanen et al., 2007*; *Sundin et al., 2007*). Thus, for the future translational purposes, we aimed to assess the ability of unconditioned urothelial cell media to substitute cell line-derived CM and pHPL as alternative to FBS, to induce ADSCs differentiation towards urothelial cells.

Our results indicate that although ADSCs are capable of expressing early differentiation markers of urothelial like cells such as CK-18 and CK-19 when cultured in CM, this expression is minimal and could not be detected at the cellular level as observed by immunostaining. On the other hand, terminal differentiation phenotype associated with the expression of UPK-2 is only achievable in the presence of CM. This confirms the presence of certain signaling factors in the CM that act in a paracrine manner and are essential for the induction process. *Zhang et al. (2013)* reported an increased level of expression of a panel of cytokines and growth factors following 12 h of induction of ADSC into urothelial cells. However, after 21 days of induction, the levels normalized slowly until they reached a level similar to the baseline level seen before the induction process (*Zhang et al., 2013*). Thus, a critical timing window for the differentiation process is crucial during the early and intermediate stages of differentiation.

Since UCM by itself lead to a minimal increase in the urothelial markers, an addition of growth factors and cytokines during this critical timing window might increase the differentiation potential. In a previous study, the addition of FGF-10 to Warton jelly-derived MSC induced their differentiation towards urothelial cells, as evident by the co-expression of CK-8 and UPK-III (*Chung & Koh, 2013*). In a similar manner, discovering more of such inductive factors paves the way for the generation of a defined differentiation system.

Here we report increased levels of EGF, VEGF, and PDGF-BB growth factors when cells were cultured in urothelial cell-derived CM. Several other studies also showed elevated levels of secretion when either ADSCs or BMSCs are cultured under the same conditions (*Tian et al., 2010*; *Shi et al., 2012*; *Zhang et al., 2014*). EGF plays a crucial role in inducing

early ESC differentiation towards the endodermal lineage (*Cras-Méneur et al., 2001*; *Kumar et al., 2014*). However, induction of terminal urothelial differentiation in vitro with PPAR γ activators requires the inhibition of EGF pathway (*Varley et al., 2004a*; *Varley et al., 2004b*; *Varley et al., 2006*). Thus, the requirement of EGF might be essential during the early stages of induction, to induce a basal urothelial phenotype and it should be followed by inhibition stage in order to produce the terminal superficial urothelial phenotype.

It has been indicated that PDGF can induce proliferation and differentiation of cells originating from all three germ layers including lung, microvilli, gastrointestinal and endothelial cell development based on the normal PDGF signaling (*Boström, Gritli-Linde & Betsholtz, 2002*; *Karlsson et al., 2000*; *Utoh et al., 2003*; *Ding et al., 2004*; *Calver et al., 1998*). Additionally, PDGF is known to be secreted by endothelial and epithelial cells and it regulates the proliferation and differentiation of neighboring smooth muscle cells (*Boström, Gritli-Linde & Betsholtz, 2002*; *Barkauskas et al., 2013*).

On the other hand, VEGF mediates MSC differentiation towards the endothelial cell lineage via the Rho/MRTF-A pathway (*Wang et al., 2013*). Although the major role of VEGF is played during vasculoneogensis and cardiac development, it also contributes significantly to the development of organs of endodermal origin including liver, lungs and pancreas (*Carmeliet et al., 1996*; *Giordano et al., 2001*; *Lammert, Cleaver & Melton, 2001*; *Matsumoto et al., 2001*; *Compernolle et al., 2002*).

Even though the role of these growth factors during the differentiation, development and organogenesis is well-established, a need arises to characterize the extent of involvement of each of these factors in the induction of urothelial cell differentiation. The general consensus on the mechanism of action of conditioned medium is the presence of undefined soluble factors released by the differentiated cells as a way of intercellular communications. These biologically active factors act in a paracrine manner and trigger an internal signaling pathways in MSCs, directing them to differentiate towards a certain lineage (*Alves da Silva et al., 2015*; *Zhang et al., 2014*). These differentiated cells in turn secrete bioactive factors, which act in an autocrine and paracrine manner to maintain the differentiated state of the cells, as well as inducing the differentiation of neighboring cells (*Alves da Silva et al., 2015*; *Zhang et al., 2014*; *Shi et al., 2012*).

In terms of serum choice, we found that pHPL is not an efficient alternative for urothelial differentiation at a concentration of 5% and 2.5%. Substituting FBS with pHPL during the differentiation reduces the expression of urothelial related markers.

Several studies reported an increase in the differentiation potential of several types of MSCs towards the osteogenic, myofibroblastic, cardiomyogenic and adipogenic cell lineages in the presence of pHPL as a serum substitute (*Karadjian et al., 2020*; *Samuel et al., 2016*; *Chignon-Sicard et al., 2017*; *Homayouni Moghadam, Tayebi & Barzegar, 2016*). On the other hand, pHPL decreased the ability of certain types of stem cells to differentiate into osteoblasts, chondrocytes and adipocytes (*Gruber et al., 2004*; *Lee et al., 2014*; *Chignon-Sicard et al., 2017*). Thus, the response elicited towards the presence of pHPL in the culture media is affected by the stem cell type, the differentiation lineage, and the percentage of pHPL used in the culture media. MSC including ADSCs have an increased proliferative capacity in the presence of pHPL (*Karadjian et al., 2020*; *Naaijkens et al., 2012*), (*Trojahn*

![PeerJ]

*Kølle et al., 2013*). As urothelial cells differentiate from the basal layer towards the superficial layer, they lose their ability to regenerate. Thus, the molecular circuitry governing differentiation depends on a skew towards differentiation by reducing proliferation, which probably could not be achieved in the presence of pHPL.

In conclusion, the induction of ADSCs towards the urothelial phenotype requires the presence of both CM and FBS. Substituting CM with UCM and FBS with pHPL significantly impacts the differentiation process. The levels of EGF, VEGF, and PDGF have increased during the differentiation process and thus might play an essential role in defining the mechanism of action of CM directed differentiation. Additionally, we found that pHPL at 5% and 2.5% concentrations negatively influenced the differentiation process. This might be explained by a skew in cellular circuitry towards the proliferation rather than the differentiation process. Changing MSCs microenvironment to accommodate the urothelial cells microenvironment is a basic requirement for any successful differentiation protocol. This can be achieved by the addition of defined soluble factors, direct or indirect co-culture with differentiated cells, the use of conditioned media or direct implantation in vivo. Urothelial cell-derived CM represents a practical and efficient way for ADSCs differentiation into urothelial lineage. However, a demand arises to formulate a defined media that efficiently induces MSC differentiation towards the urothelial lineage for clinical purposes.

### Funding
This work was supported by the Deanship of Research at the University of Jordan and the Jordan University for Science and Technology. The funders had no role in study design, data collection and analysis, decision to publish, or preparation of the manuscript.

### Grant Disclosures
The following grant information was disclosed by the authors:
University of Jordan and the Jordan University for Science and Technology.

### Competing Interests
The authors declare there are no competing interests.

### Author Contributions
- Ban Alkurdi conceived and designed the experiments, performed the experiments, analyzed the data, prepared figures and/or tables, authored or reviewed drafts of the paper, and approved the final draft.
- Nidaa A. Ababneh analyzed the data, prepared figures and/or tables, authored or reviewed drafts of the paper, and approved the final draft.
- Nizar Abuharfeil performed the experiments, prepared figures and/or tables, and approved the final draft.
- Saddam Al Demour analyzed the data, authored or reviewed drafts of the paper, and approved the final draft.

- Abdalla S. Awidi conceived and designed the experiments, authored or reviewed drafts of the paper, and approved the final draft.

## Human Ethics

The following information was supplied relating to ethical approvals (i.e., approving body and any reference numbers):

Ethics Committee at the King Abdullah Hospital, School of Medicine, Jordan University of Science and Technology approved this study (IRB No: IRB/7/2019).

## Data Availability

Raw data are available in the Supplemental Files.

## Supplemental Information

Supplemental information for this article can be found online at http://dx.doi.org/10.7717/peerj.10890#supplemental-information.

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
