# Peer review of "Use of conditioned media (CM) and xeno-free serum substitute on human adipose-derived stem cells (ADSCs) differentiation into urothelial-like cells"

_PeerJ, doi:10.7717/peerj.10890_

## Round 0.1 · original submission · Major Revisions

Please carefully address reviewers' comments and submit your revised manuscript.

Figure quality needs to be improved and text has minor changes that were pointed out by the reviewers.

·

Basic reporting

The article subject is relevant but significant changes need to be made to improve its quality.

The English language needs to be improved. The authors sometimes use informal words or sentences. Some examples: lines 65 (“What facilitates...”) and 90 (“Until now...”). Even, an orthografic error was found at line 93: word trans-differentiation.

I could no open the Supplemental files bacause of its extension .sav.

The references need to be actualized to recent findings and either standardized. See lines 102/103 and 316 for example.
The color of the text needs to be reviewed. Eg., lines 280, 337 and 343 (references).

All the figures need to be reviewed. The size of the letters are not uniform, some figures are distorted. Fluorescent images need to be with a higher quality. The legends are doubled.

It’s necessary to put the asteristic symbol (*) in the grafics whether the comparison was statistic relevant. I suggest to add in all legends the number of experiments performed.

Some ponctual alterations in the figures:

Figure 1: I suggest do divide this figure in two. The B section could be another figure and the figures letters in the following way: A, B, C... Also, its important top ut arrows or another symbol to point the structures you referenced., as lipid droplets.
Figure 2: Graphic distorted. Compare CM versus UCM data set.
Figure 3: The letter A is hidden. I suggest to include at least images of UCM-FBS condition to verify if is FBS or CM the protagonist eliciting ADSCs urothelial differentiation. Improve fluorescent images quality.
Figure 4: Graphic distorted. Compare CM versus UCM data set.

Experimental design

The research question and aim of the study and, consequently, the work originality are not clear. Did the authors want to verify whether ADSCs urothelial differentiation depended on the use of UCM and CM or FBS and pHPL? Or both?

Explain why did the cells were used at passage 3-5. Also, explain why did the analyses were performed after 14 days of differentiation (ADSCs into adipocytes or urothelial cells) or 21 days (ADSCs into osteoblasts).

I suggest to include a Materials and Reagents topic in the beginning of Method section.

Validity of the findings

The authors have not proved that the CM is the protagonist eliciting ADSCs urothelial differentiation.

According to the results, FBS seems to be the major contributor to this process. I suggest to include a CM condition or CM-Free condition (without FBS or pHPL) in the experiments, aiming to evaluate whether the CM has a predominant role promoting urethelial differentiation or if is the FBS. Even, I suggest to compare the data set of CM with UCM to support this issue.

In conclusion, the entire article needs to be reviwed extensively. The aim and results need to be more clear and robust. The discussion and introduction need to be atualized and modief depending on changes in aim and results.

·

Basic reporting

The paper aimed to investigate the role of unconditioned urothelial cell media (UCM) and pooled human platelet lysate (pHPL) as alternatives to conditioned urothelial cell media (CM) and fetal bovine serum (FBS) to induce differentiation of adipose-derived stem cells (ADSCs) into urothelial cells.

The search for alternatives to undefined culture medium is of great relevance, especially when dealing with cells with therapeutic potential, since undefined factors present in cells-derived conditioned media or FBS can affect cellular biology and physiology in uncountable ways. Besides, use of xenogenic products are also hampered by ethical concerns.

The manuscript is well-written in most of its parts (see exception below), but a couple of references would help to contextualize the problem and improve the discussion. Kolle et al. 2013 (doi:10.1016/j.jcyt.2013.01.217) showed that adipose-derived stem cells cultivated in serum-free medium supplemented with pHPL present normal chromosomal stability and differentiation capacity toward adipogenic, osteogenic or chondrogenic lineages. Besides, treatment of cells with pHPL reduced the cell population doubling time. Karadjian et al. 2020 (doi:10.3390/cells9040918) showed that the proliferation rate of bone-marrow-derived mesenchymal stromal cell (BMSC) cultured in medium supplemented with pHPL exceeded the observed in medium containing FBS. Also, differentiation of those cells toward osteogenic lineage was not negatively affected by pHPL. Thus, it’s tempting to analyse how differentiation of ADSCs into urothelial cell lineage would be affected by pHPL.

Raw data has been included, although I was unable to open the files with the provided extensions.

Microscopy images are nicely presented overall, but improvements can be made in the scale bars. All graphs shown are in very poor quality (thin lines, narrowed letters, no statistical data in the graphs, low resolution). Figure subtitles should include the number of experiments and/or replicates performed.

Experimental design

The most important issue concerns to the methods used for obtaining the conditioned media and inducing cell differentiation, which are confusingly and/or incompletely described.

Authors state that “SV-HUC-1 cells were cultured in F-12K medium supplemented with 10% FBS and 1% streptomycin and penicillin. Conditioned medium derived from SV-HUC1 was collected and used to induce the ADSCs differentiation towards the urothelial-like cell lineage”. If that description is correct and complete, the CM used in the experiments contains FBS by itself, what would invalidate all data concerning the effect of pHPL alone in cell differentiation.

Authors present data showing that expression of some urothelial cell markers (cytokeratin-18, cytokeratin-19 and uroplakin-2) is increased only in the presence of FBS, either using CM or UCM, as seen either by immunofluorescence and qRT-PCR. Additionally, production of the main growth factors released by urothelial cells in culture (EGF, PDGF and VEGF) is increased following treatment with CM or, to a lesser extent, UCM, supplemented with either FBS or pHPL. Use of pHPL had little or no effect on differentiation of cells. To validate those findings, a serum-free CM must be used.

Validity of the findings

Included in the above comment.

Additional comments

As authors conclude themselves, since UCM doesn’t affect expression of urothelial markers, it will be necessary to add growth factors, cytokines or others molecules to increase differentiation potential of the culture medium. Further studies will help to clarify the question of what combination of molecules are necessary and sufficient to direct differentiation of a stem-cell toward a urothelial cell without the need of xenogeneic components? By now, we still need to use the undefined conditioned medium with FBS to produce those cells.

---

## Round 0.2 · Minor Revisions

Dear Dr. Alkurdi, thank you for resubmitting the revised version of your manuscript. By addressing the points raised by the reviewers, the document is improved and may be acceptable for publication at PeerJ.

However both the reviewers consider crucial for this paper that CM-Free condition (without FBS or pHPL) is also tested as an additional control. I believe that this remained as a major issue to draw further conclusions based on the present work.

·

Basic reporting

The English language was improved. The prior cited orthographic error has not been adjusted yet (word trans-differentiation – line 95 (more recent revision). Even, I suggest to rewiew the word immunofluorescence along the manuscript.

The figures and grafics are with a better quality and presentation. The immunofluorescence images as well.
Minor changes:
Figure 2 (Legend): I suggest to change the form that you write the figures letters when mentioning only two of them. Instead of writing B-D, for example, it would be better to write B&D or B and D. Because it seems to be B until D.
Figure 4: The description of each image on the left side need to be up adjusted a little up.
Standardize the graphics size.
Figure 3 (Text): I suggest adding to the sentences (lines 241 and 243) which conditions were been compared, as it was indicated at line 244.

Experimental design

The research question and aim of the study are more evidente along the paper. I just suggest to add in the conclusion section of the Abstract that “Efficient ADSCs urothelial differentiation is dependent on the use of CM AND FBS”.

Validity of the findings

I still believe that using a CM-Free condition (without FBS or pHPL) in the experiments, would be better to validate these findings. However, I understand the limitations.
However, by making the aim of the work clearer together with the comparison of the CM and UCM data set, it helped to strengthen its results.

In conclusion, the paper as a whole have improved. The changes suggested by the editor and reviewers provided insights to the authors organize the main question of the work and enrich the manuscript

·

Basic reporting

No comment.

Experimental design

As I explained in the first review, the main flaw of the work is in the methods used to obtain the conditioned medium and induce cell differentiation.

The authors use FBS in either unconditioned and conditioned media to treat cells, and than they compare the effects of these media and conclude that "induction of ADSCs towards the urothelial phenotype requires the presence of both conditioned media and FBS", which in fact has has not been proven, since FBS was present in all conditions. The authors argue that "any effect coming from FBS would be minimum", but a concentration of 2% FBS (concentration used by the authors in the 'non-serum' condition) can impact various biological processes, such as those shown by the authors (regulation of expression of cell markers and production of growth factors).

Validity of the findings

In my opinion, the data presented by the authors cannot be validated if additional experiments are not carried out using a conditioned medium prepared without FBS.

Additional comments

No comment.

---

## Round 0.3 · Minor Revisions

The manuscript was improved after two rounds of independent peer review. However some minor issues should be addressed before paper is acceptable for publication.

1) What is the protein concentration of the pHPL?
2) Was there an attempt to use higher concentrations of pHPL? Since FBS was used at a 10% concentration, why didn't authors tried the same concentration of pHPL? Are there other supplements that could (or have been) tested in substitution such as B27, etc?
3) What is the role of Wnt/b-catenin signaling pathway on urothelial cell differentiation and what how is it related to proliferative/regenerative potential?

---

## Round 0.4 · accepted · Accept

Dear Dr. Awidi,

I'm pleased to inform you that your manuscript is acceptable for publication at PeerJ.

Thank you for submitting your work and addressing the issues raised by the reviewers, which I believe improved the manuscript.

Best wishes,